# Activation of OSM-STAT3 Epigenetically Regulates Tumor-Promoting Transcriptional Programs in Cervical Cancer

**DOI:** 10.3390/cancers14246090

**Published:** 2022-12-10

**Authors:** Junho Noh, Chaelin You, Keunsoo Kang, Kyuho Kang

**Affiliations:** 1Department of Biological Sciences and Biotechnology, Chungbuk National University, Cheongju 28644, Republic of Korea; 2Department of Microbiology, College of Science & Technology, Dankook University, Cheonan 31116, Republic of Korea

**Keywords:** cervical cancer, STAT3, Oncostatin M, transcriptome, epigenome, chromatin remodeling

## Abstract

**Simple Summary:**

Oncostatin M (OSM) induces Signal Transducer and Transcription 3 (STAT3) activation, exacerbating cervical cancer. The molecular mechanism by which the OSM-STAT3 axis regulates tumor-progression-related genes in cervical cancer is not well understood. Our transcriptomic and epigenomic analysis showed that OSM-induced hypoxia, wound healing, and angiogenesis genes were significantly inhibited by SD-36, the STAT3-selective degrader. Open chromatin regions of the genes that were OSM-STAT3-regulated were consistently enriched for STAT3 binding motifs. Higher levels of OSM-regulated genes’ expression were significantly linked to a poor prognosis in patients with cervical cancer. These findings show that the OSM-STAT3 signaling pathway regulates important transcriptomic programs via epigenetic modifications and that selective STAT3 inhibition may represent a novel therapeutic approach for patients with advanced cervical cancer.

**Abstract:**

Despite improvements in preventative strategies, such as regular screenings with Pap tests and human papillomavirus (HPV) tests as well as HPV vaccinations, effective treatment for advanced cervical cancer remains poor. Deregulation of STAT3 is an oncogenic factor that promotes tumorigenesis and epithelial-to-mesenchymal transition (EMT) in various cancers. Oncostatin M (OSM), a pleiotropic cytokine, induces STAT3 activation, exacerbating cervical cancer. However, the mechanism by which the OSM-STAT3 axis epigenetically regulates tumor-progression-related genes in cervical cancer is not well understood. Here, we show that OSM-mediated STAT3 activation promotes pro-tumorigenic gene expression programs, with chromatin remodeling in cervical cancer. Reanalysis of scRNA-seq data performed in cervical cancer uncovered an interaction between the oncostatin M receptor (OSMR) on tumor cells and OSM induced by tumor-associated macrophages (TAMs). Our gene expression profiling (bulk RNA-seq) shows that OSM-induced genes were involved in hypoxia, wound healing, and angiogenesis, which were significantly inhibited by SD-36, a STAT3-selective degrader. Additionally, ATAC-seq experiments revealed that STAT3 binding motifs were preferentially enriched in open chromatin regions of the OSM-STAT3-regulated genes. Among the 50 candidate genes that were regulated epigenetically through the OSM-STAT3 axis, we found that the expression levels of *NDRG1, HK2, PLOD2*, and *NPC1* were significantly correlated with those of *OSMR* and *STAT3* in three independent cervical cancer cohorts. Also, higher expression levels of these genes are significantly associated with poor prognosis in cervical cancer patients. Collectively, our findings demonstrate that the OSM-STAT3 signaling pathway regulates crucial transcriptomic programs through epigenetic changes and that selective inhibition of STAT3 may be a novel therapeutic strategy for patients with advanced cervical cancer.

## 1. Introduction

Cervical cancer is the fourth most common cause of cancer deaths in women worldwide [1,2]. The most common histological subtypes are squamous cell carcinoma and adenocarcinoma, accounting for approximately 70% and 20% of cases, respectively [3]. The main cause of cervical cancer is human papillomavirus (HPV), which causes cervical intraepithelial neoplasia (CINs) through epithelial transformation [4,5,6]. This is a precancerous stage that increases in severity over 20 to 30 years and develops into invasive cervical cancer [7]. Because of these characteristics, cervical cancer can be prevented by routine introduction of appropriate vaccination and/or effective screening programs prior to HPV infection [4]. Nevertheless, two or three generations of women who have exceeded the target (adolescent) age and those who have already been infected cannot be vaccinated [8]. In the early stages of cervical cancer, surgical resection or radiation therapy is an effective treatment, while cisplatin-based concurrent chemoradiation is considered as primary treatment for locally advanced cervical cancer (LACC) [9]. However, woman diagnosed with recurrent or metastatic cervical cancer have limited therapeutic options [10], suggesting the need for novel treatment strategies.

Dysregulated transcription factor activity in various cancers impedes treatment by inducing immune evasion, drug resistance, and EMT [11]. Recently, as a new therapeutic strategy, inhibitors targeting transcription factors were developed; clinical trials for these new drugs are ongoing [11,12,13]. STAT3 is a transcription factor that transduces cytokine and growth-factor-related signals to the nucleus with transient activity in normal cells [14]. Because STAT3 promotes the expression of genes involved in proliferation, survival, angiogenesis, immune evasion, and metastasis [15,16,17,18], the degree of STAT3 expression is associated with the prognosis of patients in various cancers. STAT3 activity is induced primarily by IL-6 family cytokines (IL-6, OSM, LIF, CT-1, and CNTF) secreted by immune cells, epithelial cells, and stromal cells [19,20,21]. Each cytokine binds to a specific receptor (IL-6R, OSMR, LIFR, or CNTF-1R), typically using glycoprotein 130 (gp130; also known as IL6ST) to recruit the Janus kinase (JAK) and induce phosphorylation of STAT3 [22]. Thus, the JAK-STAT3 signaling pathway is activated through the IL-6 family of ligand–receptor interactions to induce tumor progression [19,23]. However, STAT3 activators and their roles in different cancers are not fully understood.

OSM is a cytokine that interacts with OSMR as a ligand to induce tumor promotion or inhibition. It was originally discovered as a factor that inhibits tumorigenesis through activation of the oncogene-induced senescence (OIS) program in the early stages of tumors [24,25]. The expression level of *OSMR* correlates with the reactivity of OSM, and STAT3 activation induced by the OSM–OSMR interaction is observed in various cells, including cancer and immune cells [26,27,28,29]. Interestingly, in the tumor microenvironment (TME), OSM is secreted mainly by TAMs to induce tumor transformation or metastasis [30,31]. In cervical cancer, overexpression of *OSMR* is associated with poor clinical outcomes [32,33,34,35]. Therefore, it is regarded as a therapeutic target for cervical cancer. STAT3 activation via the OSM–OSMR interaction increases EMT, sphere-forming capacity, and tumor invasiveness [33,36], but how the OSM-STAT3 signaling pathway regulates a distinct transcriptional program is still unclear.

In this study, we investigated the mechanism of gene regulation mediated by OSM-STAT3 activity using transcriptomic and epigenomic approaches. To resolve heterogeneity of cervical cancer cells, we reanalyzed scRNA-seq data from a cervical cancer patient and discovered the interaction between OSMR on tumor cells and OSM from macrophages. We also found that tumor-progression-related genes were expressed specific to a subpopulation of tumor cells with a high expression of *OSMR* and that STAT3 activity induced by OSM was inhibited by SD-36, a STAT3 degrader, in cervical cancer cell lines [37]. Furthermore, we demonstrated that the OSM-STAT3 axis regulates expressions of hypoxia-related genes, using RNA-seq and ATAC-seq. Intriguingly, the degree of expression levels of these genes was associated with a poor prognosis for cervical cancer patients. Our findings suggest that the OSM-STAT3 axis regulates key transcriptional and epigenetic programs and that the inhibition of STAT3 may be a novel treatment for cervical cancer patients.

## 2. Materials and Methods

### 2.1. Single-Cell RNA-Sequencing Data Analysis

Previously published scRNA-seq data from normal and tumor tissue in a cervical cancer patient [38] were used to identify cell populations with OSM–OSMR interaction and relatively upregulated *OSMR*. The Seurat package [39] was used for the reanalysis of scRNA-seq data. The ratio of nFeature_RNA > 200 and mitochondria < 20% were set as thresholds to filter cells. A uniform manifold approximation and projection (UMAP) algorithm was utilized for data dimension reduction. The integrated data were clustered with 9790 cells from normal tissue and 11,521 cells from tumor tissue. Tumor tissue cells in the scRNA-seq dataset were classified via marker genes into the following lineages: epithelial cells (marked with *EPCAM* and *CDH1*), lymphocytes (marked with *GZMB* and *CD8A*), macrophages (marked with *CD163* and *MSR1*). The epithelial compartment was then classified into EP1, EP2, EP3, and EP4 based on their regulatory pathways. The “FindMarkers” command was used to obtain differentially expressed marker genes (using adjusted *p*-value < 0.05, log2FC > 0.3). Gene ontology (GO) analysis was performed with Metascape (https://metascape.org/ (accessed on 22 September 2022)).

### 2.2. Cell-Cell Interaction Analysis

CellPhoneDB [40] is a curated repository of interactions between ligands and receptors along with molecular subunit architecture information. These details are integrated in a statistical framework to infer cell–cell communication networks in single-cell transcriptomic data. To identify the cell interaction and specific pathways involved in the cervix tumor microenvironment (TME), we used the CellPhoneDB package. Ligand–receptor interactions were then identified using the “cellphonedb method statistical_analysis” command with default parameters. The interactions were visualized using “cellphonedb plot heatmap_plot” and “cellphonedb plot dot_plot”.

### 2.3. The Human Protein Atlas

The Human Protein Atlas (https://www.proteinatlas.org/ (accessed on 4 October 2022)) provided a large quantity of transcriptomic and proteomic data in specific tissues of the human body. Immunohistochemical (IHC) protein expression of OSMR, NDRG1, HK2, PLOD2, and NPC1 in normal and cancerous cervical tissues in our study was downloaded from Human Protein Atlas.

### 2.4. Cell Cuture

Human cervical cancer cell lines HeLaS3 and SiHa were cultured in RPMI 1640 (No. 11875-119; Gibco, Grand Island, NY, USA) and DMEM (No. 11965-092; Gibco) supplemented with 10% fetal bovine serum (FBS; No. 16000-044; Gibco) and 1% penicillin–streptomycin (No. 15070-063; Gibco). All cell lines were routinely cultured at 37 °C in a humidified 5% CO_2_ incubator.

### 2.5. Flow Cytometry

The HeLaS3 and SiHa cell lines (2 × 10^5^ cells) were treated with OSM (No. 30010T, PeproTech, Rocky Hill, NJ, USA) and SD-36 (No. HY129602, MedCheExpress, Monmouth Junction, NJ, USA). After 24 h, the cells were harvested with TrypLE Express (No. 12604013; Gibco). To detect intracellular STAT3 and pSTAT3, the cells were fixed in Fixation buffer solution (No. 420801; BioLegend, San Diego, CA, USA) at 37 °C for 15 min, and permeabilized in True-Phos perm buffer (No. 425401; BioLegend) at −20 °C for 1 h. The cells were then washed with cell-staining buffer (No. 420201; BioLegend) and labeled with STAT3, pSTAT3 (Y705), and pSTAT3 (S727) antibodies (No. 371804, 651010, and 698914, respectively; BioLegend) at room temperature for 30 min. The stained cells were analyzed on a CytoFLEX (Beckman Coulter, Brea, CA, USA) and the files were plotted with FlowJo v.10.7 (FlowJo LLC, Ashland, CO, USA).

### 2.6. RNA Extraction and Real-Time Quantitative PCR

Total RNA was extracted with Ribospin II (GeneAll Biotechnology, Seoul, Korea), and 0.5 μg total RNA was reverse transcribed with a RevertAid First Strand cDNA Synthesis kit (Thermo Fisher Scientific). Real-time qPCR was performed with TOPreal qPCR 2X PreMIX (SYBR Green with low ROX; No. RT500M; Enzynomics Co. Ltd., Daejeon, Korea). Primer sequences are listed for the following genes: *OSMR* (forward) 5’-AATGTCAGTGAAGGCATGAAAGG-3’, (reverse) 5’-GAAGGTTGTTTAGACCACCCC-3’; *SNAI1* (forward) 5’-TCGGAAGCCTAACTACAGCGA-3’, (reverse) 5’-AGATGAGCATTGGCAGCGAG-3’; *HK2* (forward) 5’-TGCCACCAGACTAAACTAGACG-3’, (reverse) 5’-CCCGTGCCCACAATGAGAC-3’. The real-time qPCR conditions were as follows: one cycle at 95 °C for 10 min followed by 50 cycles of 10 s at 95 °C for denaturation, 15 s at 60 °C for annealing, and 20 s at 72 °C for extension. The melting program was performed at 72–95 °C and a heating rate of 1 °C/45 s. Rotor-Gene Q v. 2.3.1 (Qiagen, Hilden, Germany) was used to capture and analyze the spectral data.

### 2.7. RNA-Seq Analysis

After RNA extraction, libraries for sequencing were prepared using the Illumina TruSeq RNA Stranded mRNA LT Sample Prep Kit following the manufacturer’s instructions. Total RNA was used to construct the sequencing library in an Illumina NovaSeq 6000 system (Illumina, San Diego, CA, USA) in 2 × 100 bp paired-end mode.

FASTQ files generated by sequencing were quality processed and adapter sequences removed with TrimGalore. The processed raw reads were mapped to the human genome (hg38) using STAR with default parameters [41]. HOMER was used to convert sorted reads into tag directories and quantify them with the “analyzeRepeats” script. Gene expression levels for each sample were normalized to fragments per kilobase of transcript per mapping read (FPKM). Differentially expressed genes (DEGs) were identified by DESeq2 analysis using raw read counts [42]. To obtain DEGs, HOMER’s “getDifferentialExpression” command was used with adjusted *p*-values < 0.05, log2FC > 1, and FPKM > 2. Gene ontology (GO) analysis was performed with Metascape. Gene set enrichment analysis (GSEA) was performed to determine gene set enrichment from the Hallmarks, KEGG, and Reactome collections for DEGs in MSigDB (http://www.gsea-msigdb.org/ (accessed on 22 September 2022)).

### 2.8. Omni-Assay for Transposase-Accessible Chromatin (ATAC) Protocol

OSM, SD36-treated HeLaS3 (5 × 10^4^ cells) was pretreated with 200 U/mL DNase (Worthington) for 30 min at 37 °C to remove free-floating DNA and digest DNA from dead cells. Then this medium was washed with DPBS. After cell harvesting through TrypLE Express, cell lysis was performed by adding a cold lysis buffer (48.5 μL RSB (resuspension buffer; 10 mM Tris-HCl pH 7.4, 10 mM NaCl, and 3 mM MgCl2 in water), 0.5 μL 10% NP-40, 0.5 μL 10% Tween-20, 0.5 μL 1% Digitonin). This cell lysis reaction was incubated on ice for 3 min. After lysis, 1 mL of RSB containing 0.1% Tween-20 (without NP-40 or digitonin) was added, and the tubes were inverted to mix. Nuclei were then centrifuged for 10 min at 600 r.c.f. in a pre-chilled (4 °C) fixed-angle centrifuge. The supernatant was removed with two pipetting steps, as described before, and nuclei were resuspended in 50 μL of transposition mix (25 μL 2 × TD buffer, 2.5 μL Tn5 transposase, 16.5 μL PBS, 0.5 μL 1% digitonin, 0.5 μL 10% Tween-20, and 5 μL water) by pipetting up and down six times. Transposition reactions were incubated at 37 °C for 30 min in a thermomixer with shaking at 1000 r.p.m. Reactions were cleaned up with MinElute Reaction Cleanup Kit. Reactions were pre-amplified for 5 cycles using NEBNext 2× MasterMix (with adapter primers). After qPCR amplification, the amplification profiles were manually assessed to determine the required number of additional cycles to amplify. See Buenrostro et al., 2015 for a detailed description of how to properly amplify the ATAC-seq library [43]. The final amplified ATAC samples were stored at −80 °C prior to sequencing analysis. Real-time PCR was performed for quality check.

### 2.9. ATAC-Seq Analysis

Using Bowtie2 with default parameters, sequenced reads were aligned to the reference human genome (GRCh38/hg38 assembly) and clonal reads were removed from further analysis. At least 10 million uniquely mapped reads were obtained for each condition. The “findPeaks” command from HOMER followed by “makeTagDirectory” was used to identify peaks of ATAC-seq enrichment in the background. A false discovery rate (FDR) threshold of 0.001 was used for all data sets. The total number of mapped reads for each sample was normalized to 10 million mapped reads. Differential peak analysis was performed using the command “getDifferentialPeaksReplicate.pl” with edgeR [44].

### 2.10. ATAC-Seq Data Visualization

Visualization of ATAC-seq data for all genes was visualized using “plotHeatmap”, a file integrated via “bamCoverage” and “computeMatrix” in deeptools [45], and using the Integrated Genome Viewer (IGV) for specific genes.

### 2.11. Motif Enrichment Analysis

Transcription factor binding sites were identified by de novo motif analysis of ATAC-seq peaks using “findMotifsGenome.pl” from the HOMER package.

### 2.12. Public Expression Profiles

Gene expression RNA-seq (HTSeq-FPKM GDC Hub) of the Cervical Cancer (CESC) cohort in The Cancer Genome Atlas (TCGA) (3 normal and 306 tumor samples) were downloaded from the UCSC Xena database. The expression profiles of GSE6791 (8 normal, 20 tumor samples) and GSE29670 (17 normal, 45 tumor samples) were downloaded from GEO and used as a test set for external validation. The overall survival rate according to the expression levels of *NDRG1*, *HK2*, *PLOD2*, *NPC1,* and *EGLN3* in CESC was determined using the GEPIA2 (Gene Expression Profiling Interactive Analysis 2) database [46].

### 2.13. Stactistical Analysis

All statistical analysis was performed using GraphPad Prism Version 9 (GraphPad Software, La Jolla, CA, USA) unless otherwise noted. Data are presented as mean ± standard deviation (SD) unless stated otherwise in each figure legend. Paired *t*-test and Ordinary one-way ANOVA were used where appropriate. For all statistical analyses, the expected variance was similar between the groups that were compared, and significance was accepted at the 95% confidence level (* *p* < 0.05, ** *p* < 0.01, *** *p* < 0.001, **** *p* < 0.0001).

## 3. Results

### 3.1. OSM–OSMR Interaction between Epithelial Cell and Macrophage in Cervical Cancer

Ligand–receptor interactions of the IL-6 family promote tumor growth through their underlying cellular mechanisms in various cancers [19], but which of these interactions mainly affect cervical cancer remains elusive. To identify crosstalk between differentially expressed receptors and ligands of the IL-6 family in cervical cancer, we reanalyzed scRNA-seq data (GSE168652) from a cervical cancer patient [38]. A total of 21,311 cells were detected, including 9790 cells from a single normal tissue and 11,521 cells from a tumor tissue (Figure 1a). To analyze the cervical cancer microenvironment, only cells from the tumor tissue were re-clustered and classified into three cell populations, using known marker genes for epithelial/cancer cells, lymphocytes, and macrophages (Figure 1b). We then identified cell-to-cell communication networks in the tumor tissue using CellPhoneDB, a database of all known ligand–receptor interactions between different cell types. The results indicate that epithelial/cancer cells exhibit the most interactions with macrophages (Figure 1c). Notably, the OSM–OSMR interaction between macrophages and epithelial/cancer cells is greater than other relationships between IL-6 family ligands and their receptors (Figure 1d). *OSMR* was consistently expressed mainly in epithelial/cancer cell populations, whereas the ligand *OSM* was specifically expressed in macrophages (Figure 1e). We also confirmed that the protein levels of OSMR are considerably increased in tumor tissues, compared to normal tissues, when using immunohistochemistry (IHC) data from the Human Protein Atlas (HPA) (Figure 1f). Furthermore, to investigate the association of *OSMR* expression levels with the prognosis of cervical cancer patients, we analyzed cervical cancer patient data from The Cancer Genome Atlas (TCGA) and found that *OSMR* overexpression was highly associated with adverse survival rates (Figure 1g). Overall, these results suggest that the OSM–OSMR axis is an important pathway among the IL-6 family in cervical cancer and that *OSMR* expression levels are significantly associated with a poor prognosis in cervical cancer patients.

### 3.2. High OSMR Expression Is Correlated with Tumor Progression Characteristics of Cervical Cancer

Given the heterogeneous characteristics of cervical cancer cells [38], we next investigated the expression level of the *OSMR* gene among subgroups of tumor cells. To this end, we further classified the epithelial/cancer cells of the tumor tissue (Figure 1b) into four subpopulations: EP1, EP2, EP3, and EP4 (Figure 2a). Interestingly, *OSMR* expression was highest in the EP1 cluster, followed by the EP2 cluster, and low in the EP3 and EP4 clusters (Figure 2b). Differentially expressed gene (DEG) analysis identified a total of 613 significantly altered genes in each cluster compared to the rest (log2FC > 0.3, adjusted *p* < 0.05). As expected, the *OSMR* gene was included in one of the marker genes for the EP1 cluster (Figure 2c). To investigate biological pathways specific to each cluster, we performed gene ontology (GO) analysis with DEGs that were representatively expressed in each population. Genes involved in ‘VEGFA-VEGFR2 signaling’, ‘response to wounding’, ‘response to hypoxia’, and ‘angiogenesis’, which related to tumor progression [47,48,49], were highly expressed in the EP1 cluster compared to the other clusters. On the other hand, genes associated with cell cycle regulation were preferentially expressed in the EP2 cluster, while ‘oxidative phosphorylation’- and ‘ATP metabolic process’-related genes were highly expressed in the EP3 and EP4 clusters (Figure 2d). Together, these results suggest that a particular cell population with high *OSMR* expression in heterogeneous cervical cancer cells may be involved in tumor exacerbation with impaired regulation of hypoxia, angiogenesis, wound healing, and cell cycle pathways.

### 3.3. Regulation of OSM–OSMR Signlaling by STAT3 in Cervical Cancer Cells

Given that the OSM–OSMR interaction leads to the activation of STAT3 in various cancer cells [26,27,28,29], we investigated the effect of OSM on STAT3 activity in cervical cancer. Intriguingly, we found significant and positive correlations with expression levels of the *OSM*, *OSMR,* and *STAT3* genes in cervical squamous cell carcinoma and endocervical adenocarcinoma (CESC) by analyzing the TCGA cohorts (Figure 3a). This suggests that the OSM-OSMR-STAT3 signaling pathway becomes active in cervical cancer. To confirm this OSM-STAT3 activation, we performed the following in vitro experiments using flow cytometry (Figure 3b). The OSM treatment altered the expression of both total STAT3 and pSTAT3 (Y705 and S727) proteins in cervical cancer cell lines (HeLaS3 and SiHa cells). Interestingly, upon long-term exposure, a further increase in their expression levels was observed (Figure 3c). Next, we treated inhibitors to modulate STAT3 activity. However, ruxolitinib, a JAK inhibitor [50,51], failed to inhibit pSTAT3 in HeLaS3 (Appendix A). Therefore, we treated cells with SD-36, a STAT3-specific degrader, for 24 h to check if the increased OSM signaling was regulated by STAT3 (Figure 3b). The protein levels of the total STAT3 and OSM-induced pSTAT3 were significantly reduced by SD-36 treatment in both cervical cancer cell lines (Figure 3d). These results indicate that OSM–OSMR signaling activates cervical cancer cells in a STAT3-dependent manner.

### 3.4. Gene Signatures Regulated by OSM-STAT3 Activity in Cervical Cancer

To examine how OSM-activated STAT3 alters gene expression programs, we performed a transcriptome analysis of cervical cancer cell lines using RNA-seq under the same conditions as the flow cytometry analysis (Figure 3b). We identified 429 DEGs (log2 of fold-change > 1, adjusted *p*-value < 0.05, and FPKM > 2) by comparing four different conditions. Clustering analysis showed that the gene expression patterns regulated by OSM and SD-36 were divided into two clusters: 338 genes that increased with OSM and decreased upon co-treatment with OSM and SD-36 (*Regulated by OSM-STAT3*) along with 91 genes that were down-regulated by OSM (*Down-regulated by OSM*) (Figure 4a,b). We focused on genes in the “*Regulated by OSM-STAT3*” cluster that exhibited a similar expression pattern to STAT3 (shown in Figure 3d). We then confirmed the expression changes of some of the identified marker genes (*OSMR, SNAI1*, and *HK2*) for the cluster in HeLaS3 and SiHa cervical cancer cell lines by RT-qPCR (Figure 4c). GO and gene set enrichment analysis (GSEA) showed that genes in the “*Regulated by OSM-STAT3*” cluster were highly associated with OSM and JAK-STAT signaling pathways. Notably, the genes in this cluster were enriched in pathways related to hypoxia, angiogenesis, and wound healing, which were observed in the EP1 cluster (Figure 2d). In contrast, tissue morphogenesis and muscle organ development pathways were significantly associated with genes in the “*Down-regulated by OSM*” cluster (Figure 4d,e). Next, we identified 91 genes common to the “*Regulated by OSM-STAT3*” cluster (338 genes in Figure 4a) and the EP1 cluster (613 genes in Figure 2). These genes were also associated with the OSM and IL-6 signaling pathways (Figure 4f). This comprehensive transcriptome analysis not only reveals that OSM-STAT3-signaling-related genes contribute to functions of hypoxia, wound healing, and angiogenesis, but also highlights the clinical importance of *OSMR*-overexpressing cells in cervical cancer.

### 3.5. Chromatin Remodeling by OSM-induced STAT3 Activity in Cervical Cancer

Nuclear phosphorylated STAT3 binds to regulatory elements in the genome to promote transcription while interacting with histone acetyltransferase (HAT) p300/CBP to induce chromatin remodeling [52]. We examined changes in the chromatin accessibility mediated by OSM-STAT3 signaling using an ATAC-seq approach. A total of 18,336 open chromatin regions (OCRs) that were differently enriched in four different conditions were identified (log2 of fold-change > 1 and adjusted *p*-value < 0.05). These peaks were further classified into three clusters: 9608 peaks with increased accessibility by OSM but reduced by SD-36 (R1 cluster), 3642 unchanged peaks (R2 cluster), and 5086 peaks with reduced accessibility by OSM (R3 cluster) (Figure 5a). An investigation of selected genes through the genome browser also confirmed that the chromatin-accessibility changes coincided with the alterations of nearby gene expression (Figure 5b). Next, we performed a de novo motif analysis to predict potential transcription factors (TFs) that bind to these clusters. The results indicated that AP-1-, CEBP-, ETS-, and KLF-binding motifs were significantly enriched in the R1 and R2 clusters (OSM-induced OCRs), whereas TEAD- and BORIS-binding motifs were found only in the R3 cluster. Interestingly, a known STAT3-binding motif was found only in the R1 cluster (Figure 5c). Since chromatin accessibility is closely related to gene regulation, core genes were defined by overlapping all genes identified by scRNA-seq, RNA-seq, and ATAC-seq. Approximately 45% of DEGs in the “*Regulated by OSM-STAT3*” cluster (identified by RNA-seq, Figure 4a) overlapped with genes regulated by chromatin remodeling (identified by ATAC-seq, Figure 5a). Of those, 50 candidate genes that were overlapped with genes in the EP1 cluster (identified by scRNA-seq, Figure 2), were finally obtained (Figure 5d). These genes were also involved in hypoxia, wound healing, and angiogenesis (Figure 5e). Thus, our integrative analysis revealed that 50 candidate genes involved in hypoxia, wound healing, and angiogenesis pathways are epigenetically regulated through the OSM-STAT3 signaling axis, and inhibited by the STAT3-degrader, SD-36.

### 3.6. OSM-STAT3 Gene Expression Signature Is Associated with Poor Prognosis in Cervical Cancer Patients

To evaluate the prognostic potential of the OSM-STAT3 gene signature in cervical cancer patients, we further analyzed RNA-seq data deposited in Gene Expression Omnibus (GEO). In the GSE6791 and GSE29570 cohorts (Table 1), 5765 and 927 genes were significantly upregulated in the tumor compared with normal samples (log2 of fold-change > 0.58 and adjusted *p*-value < 0.05), respectively. Of the 50 candidate genes we found, the *NDRG1*, *HK2*, *PLOD2*, *EGLN3*, *NPC1*, *SAT1*, *ITGA2*, and *SEMA4B* genes were upregulated DEGs in both cervical cancer patient cohorts (Figure 6a). In particular, the *NDRG1*, *HK2*, *PLOD2*, *EGLN3*, and *NPC1* genes showed a higher expression in tumors than in normal tissues, and cervical cancer patients overexpressing these genes had a poor prognosis compared with those who showed low expression levels of these genes (Figure 6b,c). Moreover, the protein levels of four selected genes (except *EGLN3* due to a lack of data) in cervical tissue were higher in tumor samples compared to normal samples, consistent with gene expression levels according to the Human Protein Atlas database (Figure 6d). Next, we tested the correlation between expression levels of these genes and *OSMR* as well as *STAT3* in the CESC cohort from TCGA. Among those, the expression levels of the *NDRG1*, *HK2*, *PLOD2*, and *NPC1* genes showed significant positive correlations with those of the *OSMR* and *STAT3* genes (Figure 6e). Notably, of the four genes, the *NDRG1*, *HK2*, and *PLOD2* genes belonged to the hypoxia pathway as identified in the pathway analysis of the 50 candidate genes (Figure 5d). Taken together, overexpression of hypoxia-pathway-related genes, such as *NDRG1*, *HK2*, and *PLOD2* [53,54,55,56,57], is induced by the OSM-STAT3 axis, and higher expression levels of these genes are significantly associated with poor prognosis in cervical cancer patients.

## 4. Discussion

Cervical cancer is mediated by HPV infection and has a precancerous stage that lasts about 30 years. Although it can be prevented with a vaccine or periodic screening [4,5,6,7], the invasive nature of cervical cancer makes treatment difficult [10]. Dysregulation of transcription factors (TFs) is associated with its invasiveness, but drug development targeting TFs is still challenging because they have few binding pockets to which drugs can bind [58,59]. Nevertheless, STAT3 is a major drug target that induces the transcription of genes involved in the proliferation, migration, invasion, angiogenesis, drug resistance, and immunosuppression of cancer [60]. Therefore, understanding the gene regulation mechanism mediated by STAT3 in cervical cancer is important for the development of innovative drugs that can directly or indirectly inhibit STAT3.

In this study, we discovered the OSM–OSMR interaction between TAMs and tumor cells in cervical cancer patient at the single-cell level. We also found that tumor progression-related genes were expressed specifically in a subpopulation of tumor cells with a high expression of *OSMR*. In fact, the OSM secretion from TAMs in TME has been well documented in various cancers, including cervical cancer [28,61,62,63]. Therefore, our results suggest that the OSM–OSMR axis serves as an important cell-to-cell communication network in cervical cancer. Based on our results, inhibiting the OSM–OSMR interaction among the IL-6 family would be a promising target for cervical cancer therapy. EGFR, like OSMR, is also known as a biomarker for cervical cancer and induces tumor progression and activation of STAT3 [64,65,66,67]. However, the EGF–EGFR interaction was not significant in the CellPhoneDB analysis. Although our findings are consistent with previous studies highlighting the importance of OSM–OSMR signaling in cervical cancer [32,33,36], the functions of other cytokines from various immune cells in cervical cancer need to be further investigated through more single-cell-based sequencing data in the future.

Our results showed that OSM stimulation for 24 h increased both the total STAT3 and phosphorylated STAT3 levels in the cervical cancer cell lines (HeLaS3 and SiHa). In contrast, previous studies reported that short-term exposure to OSM showed a reduction or maintenance pattern of STAT3 through negative feedback mediated by SOCS3 [68,69]. In our case, SD-36, a STAT3-specific degrader, reduces total STAT3 and STAT3 phosphorylation at Ser727. Thus, the differential regulation of STAT3 phosphorylation at Ser727 and Tyr705 by SD-36 may suggest distinct functions of STAT3 phosphorylation, such as mitochondrial STAT3 [70,71].

Our RNA-seq data analysis initially identified 338 genes that were regulated by OSM-STAT3 activity. Among these, 91 genes overlapped with the genes in the EP1 cluster, which consisted of *OSMR* high-expressing cervical cancer cells that were identified by the scRNA-seq data analysis. Various gene ontology analyses consistently revealed that these OSM-STAT3-regulated genes were significantly associated with hypoxia, angiogenesis, and wound healing, leading to tumor progression, drug resistance, and TME formation [47,48,49,72], but not cell-cycle-related pathways. To strengthen these findings further, we identified OSM-induced open chromatin regions (OCRs) that were regulated by OSM-STAT3 activity, using ATAC-seq. Among OSM-induced OCRs, 9608 regions (R1 cluster) contained significant numbers of STAT3 binding motifs and were selectively inhibited by SD-36. By analyzing scRNA-seq, RNA-seq, and ATAC-seq data collectively, we finally identified a core gene set (50 genes) that was epigenetically regulated by the OSM-OSMR-STAT3 axis. Among the 50 genes, *NDRG1, HK2, PLOD2,* and *NPC1* were significantly associated with poor prognosis of cervical cancer patients. Notably, *NDRG1*, *HK2*, and *PLOD2* were identified as hypoxia-associated genes in previous studies [53,54,55,56,57]. In fact, tumor hypoxia is a hallmark of cancer [73], particularly leading to resistance to radiation therapy, chemotherapy, and drugs [47,74,75,76]. This suggests that cervical cancer also has hypoxia-related characteristics that must be overcome [77,78,79,80]. Overall, our findings suggest that the hypoxia-related genes, including *NDRG1, HK2*, and *PLOD2*, can be controlled by inhibiting STAT3, and this strategy is a novel therapeutic option that may effectively reduce the progression of cervical cancer.

## 5. Conclusions

Our study demonstrates that transcriptional programs including hypoxia-related genes are regulated by OSM-STAT3 activity, and chromatin remodeling is also involved in this process in cervical cancer cells. The expression level of these hypoxia-related genes was significantly associated with the prognosis of cervical cancer patients and could be controlled by SD-36, a STAT3 degrader. Collectively, our findings suggest that STAT3 inhibition may be a novel therapeutic option for treating cervical cancer patients.

## Figures and Tables

**Figure 1 cancers-14-06090-f001:**
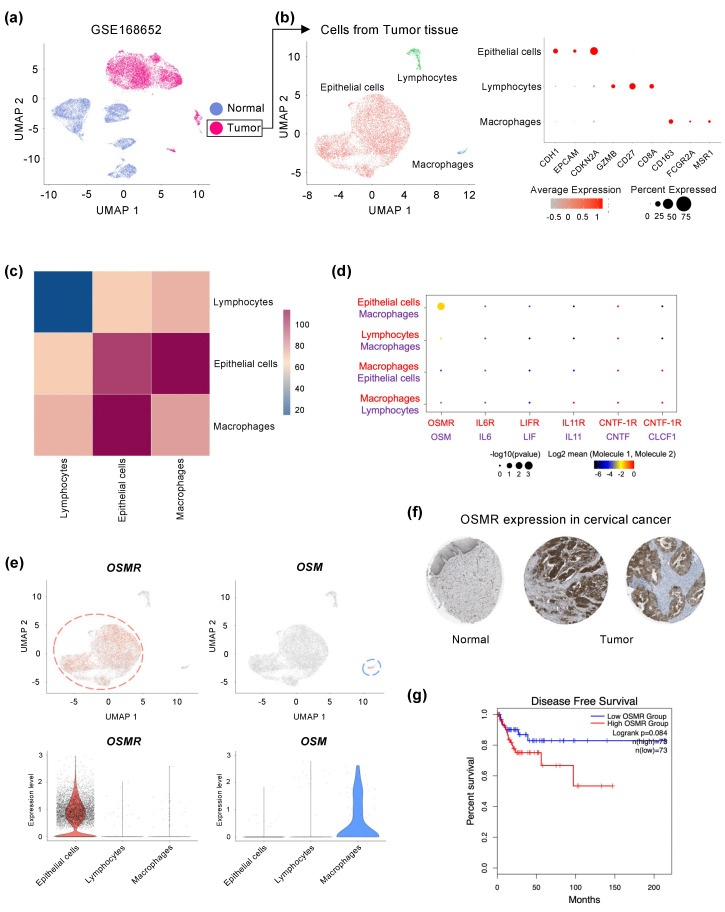
OSM–OSMR interactions are shown between epithelial cells and macrophages in the cervix tumor environment. (**a**) UMAP plots of scRNA-seq data (GSE168652) for normal and tumor tissue cells from human cervical cancer. (**b**) UMAP projection of cervix tumor tissue that were subgrouped by re-clustering analysis of scRNA-seq data. Dot plot of mean expression values of markers by cell type for epithelial cells, lymphocytes, and macrophages. (**c**) A heatmap that describes the frequency of cell-to-cell communication of the three cell types. The color of each grid represents a cell–cell interaction score. (**d**) Ligand–receptor interactions detected for the IL-6 family receptors (red) and ligands (purple) and between the different cell types. The size of the circle represents the *p*-value. The color of the circle demonstrates the mean value of cell–cell interaction. (**e**) Feature and violin plots displaying *OSMR* and *OSM* expression in cervix tumor tissue cells. (**f**) OSMR protein was upregulated in tumor tissue compared with normal cervix tissue according to The Human Protein Atlas database. (**g**) Kaplan–Meier curves showing disease-free survival for cervical SCCs with high vs low expression of *OSMR*, determined from the GEPIA2 database.

**Figure 2 cancers-14-06090-f002:**
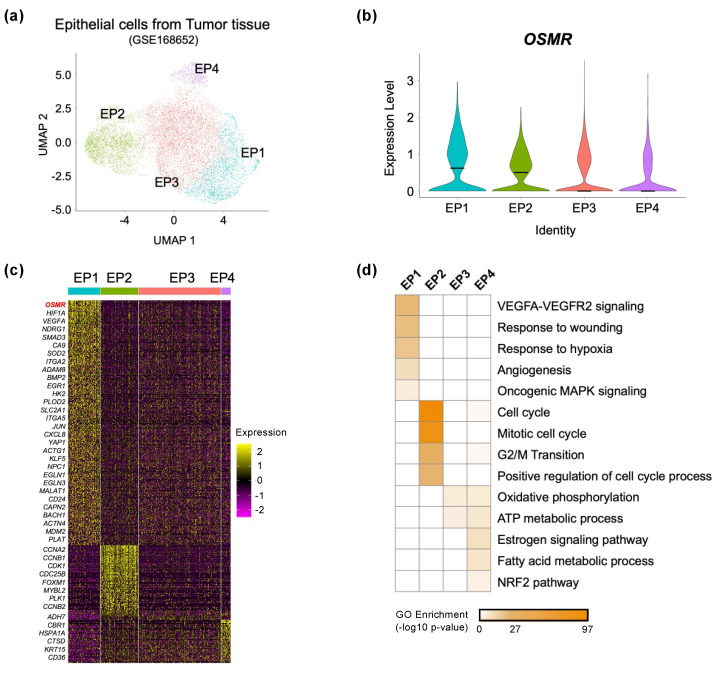
Analysis of *OSMR* expression levels for epithelial cell heterogeneity. (**a**) UMAP projection of epithelial cells that were subgrouped by re-clustering analysis of cervix tumor tissue scRNA-seq data. (**b**) Violin plot displaying *OSMR* expression in all epithelial cell clusters in cervix tumor tissue. (**c**) Heatmap shows the expression patterns of epithelial cell markers; log2FC > 0.3; adj. *p* < 0.05. (**d**) Heatmap showing the *p*-value significance of GO term enrichment for marker genes in each cluster.

**Figure 3 cancers-14-06090-f003:**
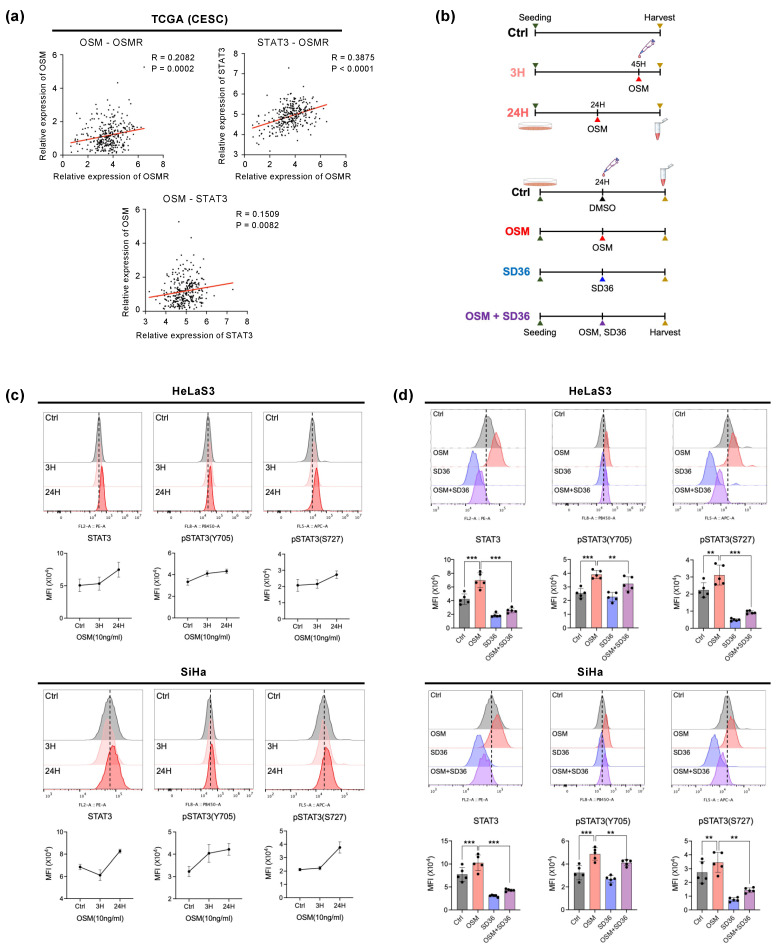
OSM-induced STAT3 activity in cervical cancer cells. (**a**) Simple linear regression analysis of expression levels of *OSM*, *OSMR*, and *STAT3* in CESC from TCGA (n = 306). (**b**) Experimental design: Upper panel; HelaS3 and SiHa were exposed to OSM (10ng/mL) for 3 or 24 h. Lower panel; HeLaS3 and SiHa underwent one of four different combinations of OSM and SD-36 (1 mM/mL) for 24 h. (**c**) Analysis of STAT3 and pSTAT3 (Y705, S727) protein expression by flow cytometry in HeLaS3 and SiHa during the time course of OSM stimulation (10 ng/mL) as indicated (Data are presented as mean ± standard error (SEM)). (**d**) Analysis of STAT3 and pSTAT3 (Y705, S727) expression by flow cytometry among four conditions; ** *p* < 0.01, *** *p* < 0.001; Paired *t*-test.

**Figure 4 cancers-14-06090-f004:**
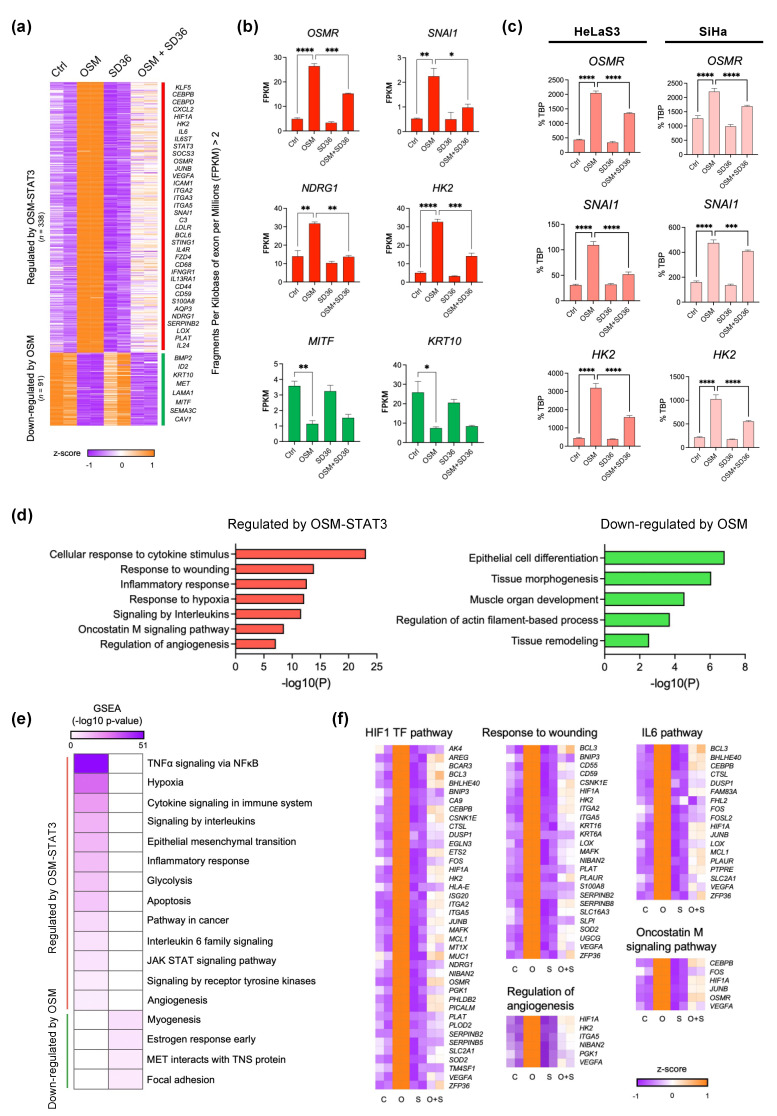
Regulation of OSM-induced transcriptome by STAT3 in cervical cancer. (**a**) Two clusters of 429 differentially expressed genes in pairwise comparisons between the four conditions; log2FC > 1; adj. *p* < 0.05; FPKM > 2. (**b**) Examples of expression of selected genes in the two clusters identified in Figure 3c. (**c**) Relative expression was normalized to internal control (TBP) and expressed relative to untreated, OSM-treated, and OSM + SD-36 treated HeLaS3 and SiHa; * *p* < 0.05, ** *p* < 0.01, *** *p* < 0.001, **** *p* < 0.0001; Ordinary one-way ANOVA. (**d**) GO analysis of DEGs in cluster “*Regulated by OSM-STAT3*” and “*Down regulated by OSM*”. (**e**) Heatmap showing the *p*-value significance of GSEA (gene sets of Hallmarks, KEGG, and Reactome) for two clusters. (**f**) Heatmap and GO term enrichment for 91 genes.

**Figure 5 cancers-14-06090-f005:**
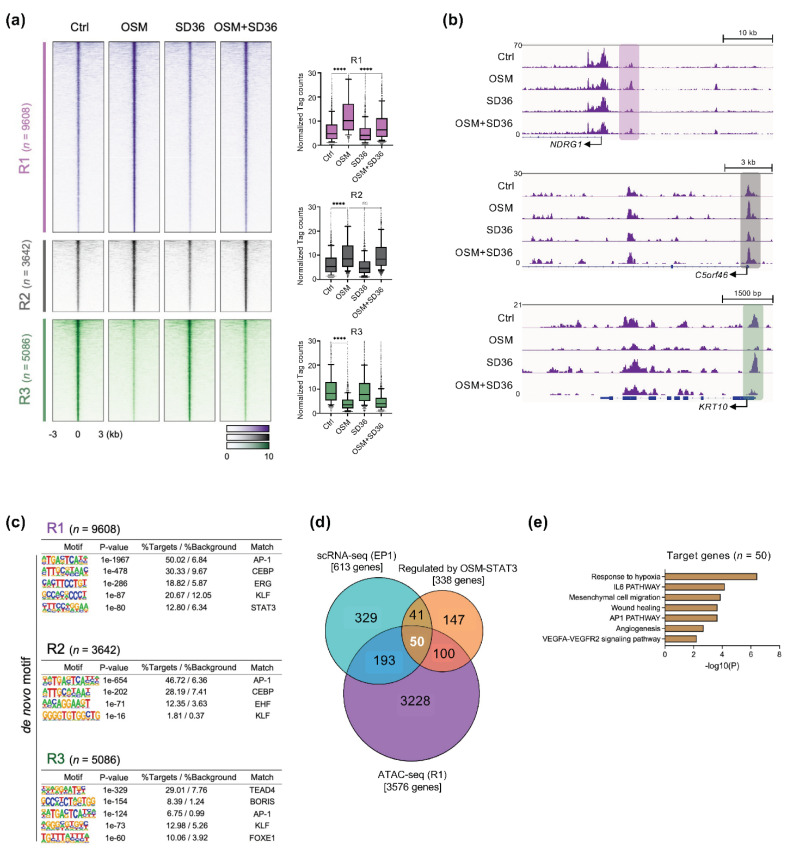
Epigenomic landscape of OSM-STAT3-activated cervical cancer cells. (**a**) Left, heatmap of open chromatin regions defined by ATAC-seq in the three clusters identified in Figure 3c. Right, the bar plot shows the normalized tag counts in each cluster; **** *p* < 0.0001; Ordinary one-way ANOVA. (**b**) Representative IGV genome browser tracks displaying normalized tag density and RNA expression level profiles in *NDRG1*, *C5orf46,* and *KRT10* under four conditions. (**c**) The most significantly enriched TF motifs were identified by de novo motif analysis using HOMER. (**d**) Venn diagram reveals intersections between DEGs in three categories including scRNA-seq, RNA-seq, and ATAC-seq. (**e**) GO analysis of 50 candidate genes.

**Figure 6 cancers-14-06090-f006:**
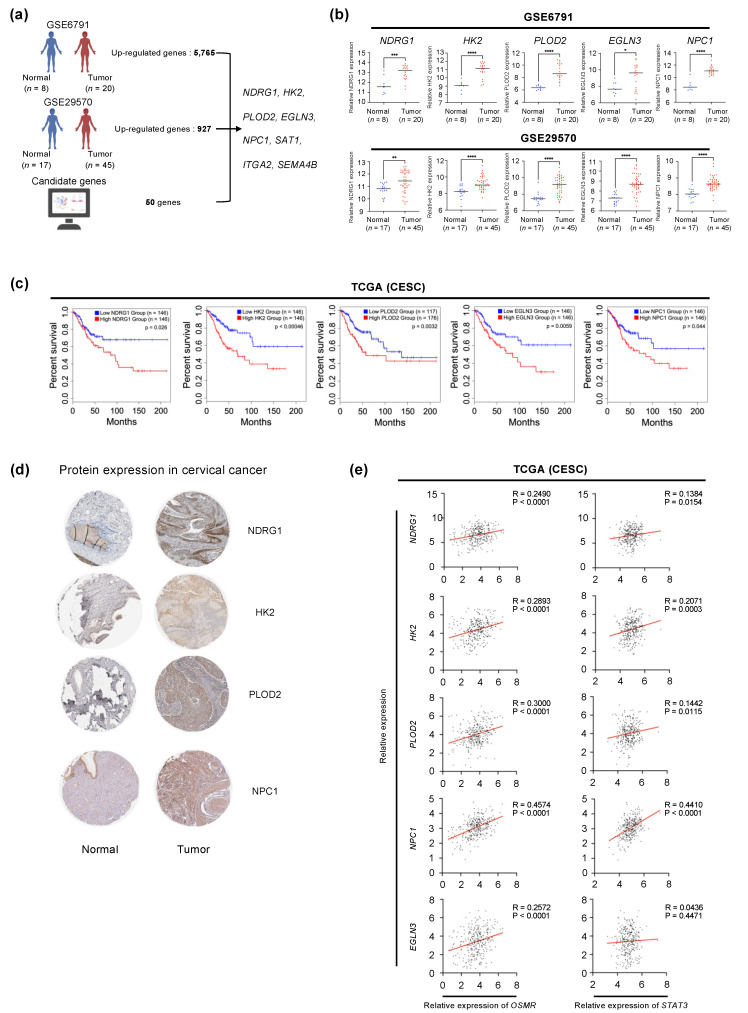
Identification of the gene signature induced by OSM-STAT3 in human cervical cancer patients. (**a**) Schematic diagram: Using publicly available microarray datasets GSE6791 and GSE29570. The schematic diagram was expressed through BioRender (https://biorender.com/ (accessed on 20 October 2022)). (**b**) Comparison of *NDRG1*, *HK2*, *PLOD2*, *EGLN3,* and *NPC1* expression levels between normal cervix tissues and cervical cancer tissues; * *p* < 0.05, ** *p* < 0.01, *** *p* < 0.001, **** *p* < 0.0001. (**c**) Kaplan–Meier curves showing overall survival for cervical SCCs with high vs low expression of five genes, as determined from GEPIA2 database. (**d**) NDRG1, HK2, PLOD2, and NPC1 proteins were upregulated in tumor tissues compared with normal cervix tissues according to the Human Protein Atlas database. (**e**) Simple linear regression analysis of expression levels of *NDRG1*, *HK2*, *PLOD2*, *NPC1,* and *EGLN3* versus *OSMR* and *STAT3* in CESC from TCGA (n = 306).

**Table 1 cancers-14-06090-t001:** Clinicopathological features of cervical cancer patients examined in this study.

Characteristics	GSE6791	GSE29570
No.	20	45
Age		
≤45	7 (35%)	19 (42%)
>45	13 (65%)	26 (58%)
Stage		
IB	16 (80%)	25 (56%)
II/III	3 (15%)	16 (36%)
IV	1 (5%)	4 (8%)
Histology		
SCC	20 (100%)	41 (91%)
ACC		3 (7%)
ASCC		1 (2%)

SCC, Squamous Cell Carcinoma. ACC, Adenocarcinoma. ASCC, Adenosquamous Cell Carcinoma.

## Data Availability

All publicly available datasets used in this study were described in the Materials and Methods. Other datasets generated in this study are available upon reasonable request to the corresponding author.

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
