# Peer review of "Activation of OSM-STAT3 Epigenetically Regulates Tumor-Promoting Transcriptional Programs in Cervical Cancer"

_cancers, 2022, doi:10.3390/cancers14246090_

Round 1

Reviewer 1 Report

The article entitled “Activation of OSM-STAT3 epigenetically regulates tumor-promoting transcriptional programs in cervical cancer”  by  Kang et al., is aimed at delineating the role of oncostatin M based regulatory role of STAT3 on the cervical cancer progressive.  Overall, the methodologies used are appropriate to assess the proposed major objectives of the manuscript. However, the following points are to be addressed to increase the understanding of the readers.

Specific points to be addressed:

Justify the selection of cell lines HeLaS3 and SiHa, why it is preferred over cell lines?

The relevance of EGFR signalling is not being thoroughly discussed in cervical cancer cells? The reference number 61 mentions about EGFR but a substantial discussion of this is missing. The author can refer the article by Muthusami et a., 2022 on EGFR signalling and cervical cancer

 The authors needs to separate the bioinformatics study and wet laboratory data in different sections for better understanding of the data

How OSM treatment was given? How the phosphorylation of STAT3 was assessed?Why the authors preferred flow cytometry for phosphorylation assessment over western blotting in cell lines? These details are to be provided in the manuscript

Minor:

Expand STAT while using for the first time in the abstract in simple summary

EMT is already abbreviated in line number 24, the usage of expanded version of EMT in line number 63 may be avoided, screen for such things for maintaining the uniformity

Author Response

Point 1: Justify the selection of cell lines HeLaS3 and SiHa, why it is preferred over cell lines?

Response 1: The idea of our project was started from STAT3 ChIP-seq data of the HeLa-S3, the only cell line of cervical cancer in the ENCODE project. To validate our results, we performed in vitro experiments using HeLa-S3 as well as the SiHa cell line, the most commonly used in cervical cancer studies.

Point 2: The relevance of EGFR signalling is not being thoroughly discussed in cervical cancer cells? The reference number 61 mentions about EGFR but a substantial discussion of this is missing. The author can refer the article by Muthusami et a., 2022 on EGFR signalling and cervical cancer.

Response 2: References have been added accordingly on page 20 [62-65]. Although EGFR is a well-known biomarker for cervical cancer and induces tumor progression and activation of STAT3, we could not find the EGF-EGFR interaction in our analysis using CellPhoneDB (scRNA-seq data).

Point 3: The authors needs to separate the bioinformatics study and wet laboratory data in different sections for better understanding of the data

Response 3: We did show our own experimental data including bioinformatic analysis (from Fig.3B to Fig.5). In addition to our NGS data, we used public scRNA-seq data (Fig.1 and Fig.2) and RNA expression data of big cohorts (Fig.6. TCGA and microarray data) in order to validate and strengthen our argument.

Point 4: How OSM treatment was given? How the phosphorylation of STAT3 was assessed? Why the authors preferred flow cytometry for phosphorylation assessment over western blotting in cell lines? These details are to be provided in the manuscript.

Response 4: We provided the detailed experimental design and conditions in Figure 3B and Figure legends. We thought it was better to use flow cytometry to measure pSTAT3 levels (using mean fluorescence intensity, MFI) quantitatively.

Minor:

  1. Expand STAT while using for the first time in the abstract in simple summary.

 - The text has been added accordingly on line 11.

  1. EMT is already abbreviated in line number 24, the usage of expanded version of EMT in line number 63 may be avoided, screen for such things for maintaining the uniformity.

- The text has been changed accordingly on line 64.

We thank the Reviewers for their helpful comments. We believe that the revised manuscript addresses the issues raised during review and has been improved. Thank you for considering the revised manuscript for publication in Cancers.

Sincerely,

Kyuho Kang, Ph.D.

Reviewer 2 Report

In this manuscript, the authors investigated the mechanism of gene regulation mediated by the OSM-STAT3 axis. The authors draw the conclusion that OSM-STAT3 signaling pathway regulates transcriptomic programs through 18 epigenetic modifications and selective STAT3 inhibition may be a novel therapeutic approach for patients with advanced cervical cancer. This is an interesting topic, while the article has some shortcomings that need to be improved. The comments are as follows:

1. In line 50, CIN is an abbreviation for cervical intraepithelial neoplasia.

2. This paper focus on the OSM-STAT3 axis, while the methods and theoretical innovations are not sufficient. The author must dig deeper into the innovation points before the article can be accepted.

3. The authors used SD-36, a STAT3-specific degrader, in the experiment to discover if the increased OSM signaling was regulated by STAT3, and found protein levels of STAT3 and pSTAT3 were significantly reduced by SD-36 treatment in both cervical cancer cell lines. The design here seems to be unreasonable to confirm OSM-STAT3 activation.

4.This paper lacks enough experimentation to demonstrate the reliability of conclusion. The author needs to do more experiments with more angles and show them in this paper.

Author Response

Point 1: In line 50, CIN is an abbreviation for cervical intraepithelial neoplasia.

Response 1: The text has been added accordingly on line 52.

Point 2: This paper focus on the OSM-STAT3 axis, while the methods and theoretical innovations are not sufficient. The author must dig deeper into the innovation points before the article can be accepted.

Response 2: First, we did show OSM-OSMR interaction between TAMs and epithelial cells at the single-cell level. Second, we did provide OSM-STAT3 mediated transcriptomic and epigenomic changes in cervical cancer cells using integrated RNA-seq and ATAC-seq approaches for the first time. Although some previous studies revealed OSMR signaling in cervical cancer, we believe that we did provide not only the systemic view of OSM-STAT3-mediated epigenetic regulation in cervical cancer but also target genes regulated by the OSM-STAT3 axis.

Point 3: The authors used SD-36, a STAT3-specific degrader, in the experiment to discover if the increased OSM signaling was regulated by STAT3, and found protein levels of STAT3 and pSTAT3 were significantly reduced by SD-36 treatment in both cervical cancer cell lines. The design here seems to be unreasonable to confirm OSM-STAT3 activation.

Response 3: The OSM-STAT3 signaling pathway is complicated to inhibit using small molecule inhibitors targeting kinase activity, such as ruxolitinib (Figure S1). We, therefore, used a PROTAC STAT3 degrader, SD-36 in our study.

Point 4: This paper lacks enough experimentation to demonstrate the reliability of conclusion. The author needs to do more experiments with more angles and show them in this paper.

Response 4: As we mentioned above in Response 2, we did our own RNA-seq and ATAC-seq experiments using cervical cancer cells. We think our results were also validated using public data sets.

We thank the Reviewers for their helpful comments. We believe that the revised manuscript addresses the issues raised during review and has been improved. Thank you for considering the revised manuscript for publication in Cancers.

Sincerely,

Kyuho Kang, Ph.D.

Reviewer 3 Report

This study showed OSM-STAT3 axis which regulates hypoxia-related genes using public TCGA data and cervical cell lines.

Major comments

1.     Although this study proved critical roles of OSM-STAT3 axis and hypoxia-related genes including NDRG1, HK2 and PLOD2, this study used only public TCGA data and cervical cell lines. So, real clinical value of this OSM-STAT3 axis and its related genes are unclear. It may be better to analysis real clinical cervical samples for evaluating clinical roles of OSM-STAT3 axis and its related genes and validate via TCGA public data.

2.     Patients’ clinical information, such as age, stage, histology, should be added

3.     Why did you choose OSM among the various IL-6 family cytokines?

Author Response

Point 1: Although this study proved critical roles of OSM-STAT3 axis and hypoxia-related genes including NDRG1, HK2 and PLOD2, this study used only public TCGA data and cervical cell lines. So, real clinical value of this OSM-STAT3 axis and its related genes are unclear. It may be better to analysis real clinical cervical samples for evaluating clinical roles of OSM-STAT3 axis and its related genes and validate via TCGA public data.

Response 1: In Figure 6, we did validate our target genes using TCGA (CESC) data as well as two independent cohorts of public data sets.

Point 2: Patients’ clinical information, such as age, stage, histology, should be added

Response 2: Table 1 has been added accordingly on page 5.

Point 3: Why did you choose OSM among the various IL-6 family cytokines?

Response 3: We did explain the reason why OSM is important in cervical cancer among IL-6 family cytokines in the Introduction part (lines 78-89). In addition, our data (Fig. 1d) showed OSM-OSMR interaction is the only significant interaction among IL-6 family cytokines in cervical cancer.

We thank the Reviewers for their helpful comments. We believe that the revised manuscript addresses the issues raised during review and has been improved. Thank you for considering the revised manuscript for publication in Cancers.

Sincerely,

Kyuho Kang, Ph.D.

Round 2

Reviewer 1 Report

The authors revised the manuscript adequately to fulfill publication 

Reviewer 2 Report

The authors have made corresponding revisions and explanations according to the review opinions,the paper can be accepted.

Reviewer 3 Report

There is no comment.